# An Efficient Asynchronous Method for Integrating Evolutionary and Gradient-based Policy Search

**Kyunghyun Lee**     **Byeong-Uk Lee**     **Ukcheol Shin**     **In So Kweon**
Korea Advanced Institute of Science and Technology (KAIST)
Daejeon, Korea
{kyunghyun.lee, byeonguk.lee, shinwc159, iskweon77}@kaist.ac.kr

## Abstract

Deep reinforcement learning (DRL) algorithms and evolution strategies (ES) have been applied to various tasks, showing excellent performances. These have the opposite properties, with DRL having good sample efficiency and poor stability, while ES being vice versa. Recently, there have been attempts to combine these algorithms, but these methods fully rely on synchronous update scheme, making it not ideal to maximize the benefits of the parallelism in ES. To solve this challenge, asynchronous update scheme was introduced, which is capable of good time-efficiency and diverse policy exploration. In this paper, we introduce an Asynchronous Evolution Strategy-Reinforcement Learning (AES-RL) that maximizes the parallel efficiency of ES and integrates it with policy gradient methods. Specifically, we propose 1) a novel framework to merge ES and DRL asynchronously and 2) various asynchronous update methods that can take all advantages of asynchronism, ES, and DRL, which are exploration and time efficiency, stability, and sample efficiency, respectively. The proposed framework and update methods are evaluated in continuous control benchmark work, showing superior performance as well as time efficiency compared to the previous methods.

## 1   Introduction

Reinforcement Learning (RL) algorithms, one major branch in policy search algorithm, were combined with deep learning and showed excellent performance in various environments, such as playing simple video games with superhuman performance [1, 2], mastering the Go [3], and solving continuous control tasks [4–6]. Evolutionary methods, another famous policy search algorithm, were applied to the parameters of deep neural network and showed compatible results as Deep Reinforcement Learning (DRL) [7].

These two branches of policy search algorithms have different properties in terms of sample efficiency and stability [8]. DRL is sample efficient, since it learns from every step of an episode, but is sensitive to hyperparameters [9–12]. Evolution Strategies (ES) are often considered as the opposite because they are relatively stable, learning from the result of the whole episode [8], yet they require much more steps in the learning process [8, 13, 14].

ES and DRL are often considered as competitive approaches in policy search [7, 15], and relatively few studies have tried to combine them [16–18]. Recently, some works tried to utilize the useful gradient information of DRL into ES directly. Evolutionary Reinforcement Learning (ERL) has an independent RL agent that is periodically injected into the population [13]. In Cross-Entropy Method-Reinforcement Learning (CEM-RL), half of the population is trained with gradient information, and new mean and variance of the population are calculated with better performing individuals [19].

In both ES and RL, parallelization methods were introduced for faster learning and stability [2, 7, 15, 20, 21]. Most of these parallelization methods take synchronous update scheme, which aligns the update schedule of every agents to the one with the longest evaluation time. This causes crucial time inefficiency because agents with shorter evaluation should wait until the whole agents finish their job.

The asynchronous method is one of the direct solutions to this problem, as one agent can start the next evaluation immediately without waiting other agents [22–25]. Another advantage of the asynchronous method is that updates occur more often than those of synchronous methods, which can encourage diverse exploration [25, 26].

In this paper, we propose a novel asynchronous framework that efficiently combines both ES and DRL, alongside with some effective asynchronous update schemes by thoroughly analyzing the property of each. The proposed framework and update schemes are evaluated on the continuous control benchmark, underlining its superior performance and time-efficiency compared to previous ERL approaches.

Our contributions include the following:

- We propose a novel asynchronous framework that efficiently combines both Evolution Strategies (ES) and Deep Reinforcement Learning algorithms.

- We introduce several asynchronous update methods for the population distribution. We thoroughly analyze all update methods and their properties. Finally, we propose the most effective asynchronous update rule.

- We demonstrate the time and sample efficiency of the proposed asynchronous method. The proposed method reduces the entire training time about 75% wall clock on the same hardware configuration. Also, the proposed method can achieve up to 20% score gain with given time steps through effective asynchronous policy searching algorithm.

## 2 Background and Related work

### 2.1 Evolution Strategies

Evolutionary Algorithm (EA) is a type of black-box optimization inspired by natural evolution, that aims to optimize a fitness value. All individuals in EA are evaluated to calculate fitness function $f$, which is similar concept to the reward in RL. Evolutionary Strategy (ES) is one of the main branches in EA that one individual remains for each generation. When the population is represented by mean and covariance matrix, these algorithms are called estimation of distribution. The most famous algorithms in this category are the Cross-Entropy Method (CEM) [27] and Covariance Matrix Adaptation Evolution Strategy (CMA-ES) [28].

In CEM, individuals are sampled based on the distribution $\mathcal{N}(\mu, \Sigma)$. All individuals are evaluated and the distribution is updated based on the fixed number of best-performing individuals. CMA-ES is similar to CEM, except it considers the "evolutionary path" that collects the direction of consecutive generations.

### 2.2 Parallel, Synchronous and Asynchronous ES

Parallelization is the most useful way to increase the learning speed of the ES and DRL, by enabling concurrent execution. Many existing studies reported improvements using parallelization [23, 26, 29–31]. For example, Salimans et al. [7] solved MuJoCo humanoid with a simple ES algorithm in 10 minutes by using 1440 parallel cores. In particular, changing from serial to synchronous parallel requires no modification on algorithmic flow, while enabling substantial improvement in execution speed.

The synchronization process is a step that updates the population with the results of all individuals for the next generation. Therefore, when the evaluation times of all individuals differ, all workers should wait until the last worker is finished [25], thereby increasing idle time. This is more critical when the number of cores increases, since it is hard to expect linear increase of time efficiency [24].

Asynchronous algorithms are introduced to address this problem. They propose a modified process to allow all individuals to update the population immediately after their evaluation, thereby reducing

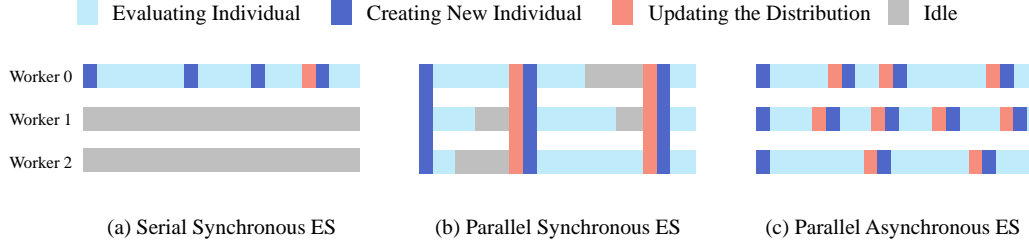

Figure 1: **Parameter Update Scheme Comparison.** (a) One worker is used in serial synchronous ES. The worker evaluate the individuals sequentially. (b) In parallel ES, all workers evaluate the individuals in parallel. However, some workers are in idle until the last worker is finished in each generation. (c) All workers update the distribution immediately after the evaluation and evaluate the next generation. Note that the number of updates per evaluation is much more frequent in (c) compare to the other methods, which encourages the exploration.

meaningless idle time. Also, more frequent updates increase the exploration in the parameter space [25, 24]. Concepts and differences are illustrated in Fig. 1.

## 2.3 Update methods in ES

The update method can be divided into two main approaches depending on how the fitness value is used: the rank-based methods and the fitness-based methods. The rank-based methods sort the fitness values of all individuals and only the rank information is used for update. The fitness-based methods use the fitness value itself. The rank-based methods are more conservative and robust to inferiors. However, it also ignores the useful information from superiors [32].

In synchronous and asynchronous perspective, rank-based methods are more suitable to synchronous methods because it needs the results of all individuals in the population. On the other hand, fitness-based methods do not have this restriction and can utilize the superior individual sufficiently. The main problem in fitness-based methods is fitness value normalizing, because the evolution pressure is directly proportional to the fitness value.

In CEM, the population is updated with the rank-based method, as

$$\mu_{t+1} = \sum_{i=1}^{K_e} \lambda_i z_i, \qquad \Sigma_{t+1} = \sum_{i=1}^{K_e} \lambda_i (z_i - \mu_t)(z_i - \mu_t)^{\mathsf{T}} + \epsilon \mathcal{I}, \qquad (1)$$

where $K_e$ is a number of elite individuals, $(z_i)_{i=1,\ldots,K_e}$ are parameters of selected individuals according to the fitness value, and $(\lambda_i)_{i=1,\ldots,K_e}$ are weights of selection for the individuals. These weights are generally either $\lambda_i = \frac{1}{K_e}$ or $\frac{\log(1+K_e)/i}{\sum_{i=1}^{K_e} \log(1+K_e)/i}$ [28]. The former is the method of giving the same weights for the selected individuals, and the latter is the method of giving higher weights according to rank.

The covariance matrix calculation is sometimes reduced to variance because the calculation complexity grows exponentially as the number of parameters increases [19]. Then, Eq. (1) is simplified as

$$\Sigma_{t+1} = \sum_{i=1}^{K_e} \lambda_i (z_i - \mu_t)^2 + \epsilon \mathcal{I}. \qquad (2)$$

## 2.4 Evolutionary Reinforcement Learnings

As mentioned in the introduction section, EA and DRL have opposite property in sample efficiency and stability. Based on these characteristics, there have been a few approaches to combine these two, taking advantages from both [13, 33, 14, 19].

The evolutionary reinforcement learning (ERL) [13] is the pioneering trial to merge two approaches: population-based evolutionary algorithm and sample efficient off-policy deep RL algorithm (DDPG) [4]. However, the sample efficiency issue remains an important problem, which triggered various enhanced version of ERL to be suggested. Collaborative Evolutionary Reinforcement Learning

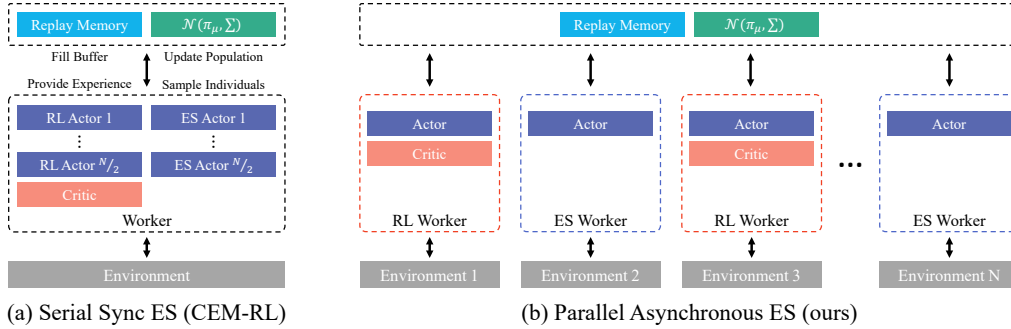

(a) Serial Sync ES (CEM-RL)    (b) Parallel Asynchronous ES (ours)

Figure 2: **Framework Comparison with Serial Synchronous (CEM-RL) and Parallel Asynchronous method (ours).** CEM-RL has one worker and evaluates all individuals sequentially. Ours has several distributed workers and evaluates individuals asynchronously

(CERL) [33] collects different time-horizons episodes by leveraging a portfolio of multiple learners to improve sample diversity and sample efficiency. Further, Proximal Distilled Evolutionary Reinforcement Learning (PDERL) [14] points out that the standard operators of GA, the base evolutionary algorithm of ERL, are destructive and cause catastrophic forgetting of the traits. All ERL variants share the same architecture, an independent RL agent. That is, an RL agent is trained along with the ES, and injected into the population periodically to leverage the RL. Otherwise, CEM-RL [19] trains the half of individuals with RL algorithm, utilizing the gradient information directly.

## 3    Methods

In this section, we present a novel asynchronous merged framework of ES and RL. Any kind of the off-policy actor-critic algorithm with a replay buffer can be applied to the RL part. Also, we present effective new update method that balances between exploration and stability. A pseudocode for the whole algorithm is described in Appendix B.

### 3.1    Asynchronous Framework

As described in Fig. 1, the parallel asynchronous update scheme is more time-efficient and encourages exploration compared to the synchronous update schemes. The framework of our baseline, CEM-RL, has half ES actors and half RL actors as shown in Fig. 2-(a). Only after all evaluations of the actors are finished, CEM-RL updates population distribution represented by the mean $\mu$ and the variance $\Sigma$. On the other hand, our proposed parallel asynchronous framework allows each worker to have an ES or RL actor and individually update the population distribution as shown in Fig. 2-(b). Also, in CEM-RL, the shared critic network learns after synchronizing step and takes time for a while. In contrast, the shared critic network continuously learns in parallel with the other actors in our framework. Based on the change of this update scheme, the overall training time reduction and active exploration can be expected.

However, there are some problems to convert the serial synchronous update scheme to the parallel asynchronous update scheme. The first problem occurs in creating new individuals. CEM-RL creates a fixed number of ES and RL actors in new generation only after all actors are terminated. It is hard to apply the method directly to the asynchronous update because the actors have different ending moments. This problem is handled in Sec. 3.2. Secondly, we need an asynchronous method to update population distribution effectively. Since each actor has its own fitness value, a novel update method to adaptively exploit this is required. This problem is handled in Sec. 3.3 and Sec. 3.4. Our overall algorithm pseudo-code is prested in Appendix B

### 3.2    RL and ES Population Control

To solve the new individual creating problem, we introduce a probability based population control method for the asynchronous scheme. This method probabilistically allocates a new individual as an ES or RL actor so that the ratio between RL and ES actor is maintained as a desired ratio. The probability of being an RL actor is represented as $p_{\text{rl}}$, and is adaptively determined by the cumulative

number of RL and ES actors, described as follows:

$$p_{\text{rl}} = \text{clip}\left(-K_{rl}\left[\frac{n_{\text{rl}}}{n_{\text{rl}} + n_{\text{es}}} - p_{\text{desired}}\right] + 0.5, 0, 1\right) \tag{3}$$

where $K_{\text{rl}}$ is a $P$ controller gain, $p_{\text{desired}}$ is the desired rate of RL agent, and $n_{\text{rl}}$ and $n_{\text{es}}$ are the total number of RL and ES agents, respectively.

### 3.3 Fitness-based Mean Update

In rank-based methods, the valid weights of individuals for updating distribution are solely based on the rank, therefore the amount of "how much better is the agent" is ignored. To address this problem, we consider fitness-based methods. Fitness-based methods use the fitness value itself rather than rank, making it available to fully utilize the superior individual. One of the early works, (1+1)-ES [34], compares the fitness values of a current and new individual. Although it is not considered as a fitness-based method, we can consider it as an extremely aggressive fitness-based method, since it moves the mean $\mu$ entirely to the new individual when the fitness value of the new one is better than the current one. For smoother update, we design the mean update algorithm as Eq. (4),

$$\mu_{t+1} = (1 - p)\mu_t + p \cdot \mathbf{z} \tag{4}$$

where $p$ is an update ratio between two parameters, and $\mathbf{z}$ is a parameter of newly evaluated individual. Regarding Eq. (4), $p$ in (1+1)-ES can be expressed as

$$p = \begin{cases} 1 & \text{if } f(\mu_t) < f(\mathbf{z}) \\ 0 & \text{if } f(\mu_t) \geq f(\mathbf{z}) \end{cases} \tag{5}$$

In order to restrain the aggressiveness of (1+1)-ES, while maintaining the capability of obtaining much information from superior individuals, we propose two novel fitness-based update methods.

**Fixed Range**   First one is the fixed range method. This method updates the distribution relative to the current fitness value. The update factor is determined by the fixed range around the fitness value of current mean. We evaluate two variants of the fixed range methods, it sets the update rate by clipped linear interpolation over a fixed range hyperparameter $r$.

$$p = p_{\text{sg}} \cdot \text{clip}\left(\frac{f(\mathbf{z}) - f(\mu_t)}{r}, -1, 1\right) \tag{6}$$

$p_{\text{sg}}$ is determined as follow,

$$p_{\text{sg}} = \begin{cases} p_{\text{positive}} & \text{if } f(\mu_t) < f(\mathbf{z}) \\ p_{\text{negative}} & \text{if } f(\mu_t) \geq f(\mathbf{z}) \end{cases} \tag{7}$$

where $p_{\text{positive}}$ and $p_{\text{negative}}$ are hyperparameters. Note that it is possible to move $\mu$ only in positive direction by setting $p_{negative}$ as 0, or to let it move backward by setting $p_{negative}$ other than 0.

Additionally, we introduce Sigmoid function for more smooth update:

$$p = p_{\text{sg}} \cdot \left[\text{Sigmoid}\left(\frac{f(\mathbf{z}) - f(\mu_t)}{r}\right)\right] \tag{8}$$

where $p_{\text{sg}}$ is the same as in Eq. (7), and $\text{Sigmoid}(x) = 1/(1 + e^{-x})$

**Fitness Baseline**   Secondly, we propose the baseline methods for more conservative updates compare to the fixed range methods. In the fixed range methods, the update ratio changes drastically with moving averages. Therefore, we apply baseline techniques to make it less optimistic, as numerous RL methods [2, 5] advantage from. There are two variants in this category: absolute baseline and relative baseline. The absolute baseline sets the baseline for the entire steps, while the relative method moves the baseline with the current mean value. We also clip the update ratio from $-1$ to $1$. The former method follows the update rule in Eq. (9),

$$p = \text{clip}\left(\frac{f(\mathbf{z}) - f_{\text{b}}}{[(f(\mu) - f_{\text{b}}) + (f(\mathbf{z}) - f_{\text{b}})]}\right) \tag{9}$$

where $f_b$ is a hyperparameter for the fitness baseline.

The relative baseline is defined by $f_{rb} = f(\mu) - f_b$, which makes an update ratio $p$ and a modified version of $p_{sg}$ to be expressed as

$$p = p_{sg} \cdot \text{clip}\left(\frac{f(\mathbf{z}) - f_{rb}}{f_b + (f(\mathbf{z}) - f_{rb})}, -1, 1\right), \qquad p_{sg} = \begin{cases} p_{positive} & \text{if } f(\mathbf{z}) \geq f_{rb} \\ p_{negative} & \text{if } f(\mathbf{z}) < f_{rb} \end{cases} \tag{10}$$

We set $p = 0$ for very low fitness value, $f(\mathbf{z}) < f_{rb} - f_b$

### 3.4   Variance Matrix Update

We update the variance matrix $\sigma I$ instead of the covariance matrix $\Sigma$ to reduce time complexity, following the assumption used in CEM-RL [19]. As described in Sec. 2.3, the variance of the synchronous method can be calculated directly from a set of individuals, but not in the asynchronous update method. Therefore, different methods such as asynchronous ES [26] and Rechenberg's 1/5th success rule [34] were introduced, where the first one utilizes pre-stored individuals, and the second one increases or decreases variance based on a success ratio $p_s$ and threshold ratio $p_{th}$. However, the former simply adapted the method of rank-based synchronous, and the latter naively controlled the variance. In this section, we propose two asynchronous variance update methods that effectively encourages exploration in proportion to the update ratio of $p$.

**Online Update with Fixed Population**   In the asynchronous update scheme, the mean and variance of all individuals are unknown, and only the values of the current population distribution and a single individual are known. Welford's online update rule [35] is an update method that can be used in this situation, described as follows:

$$\sigma_{\mathbf{t}}^2 = \sigma_{\mathbf{t-1}}^2 + \frac{(\mathbf{z} - \mu_{\mathbf{t-1}})^{\mathsf{T}}(\mathbf{z} - \mu_{\mathbf{t}}) - \sigma_{\mathbf{t-1}}^2}{n} \tag{11}$$

This equation was initially designed to save the memory space without storing the past data by calculating the new variance from the current variance. For this purpose, the original $n$ in Eq. (11) is increased each time new data comes in, to represent the total number of data. Also, past data and present data are equally valuable in the original equation. However, since the past and present individuals should not be treated equally in the ES, we need to calculate the variance by giving more weight to the values of the recent individuals. Thus, we modified $n$ in the equation to have constant value inspired by the concept of rank-based variance update method [26] which removes the oldest individual from the population. The influence of old individuals would gradually disappear with the fixed $n$.

**Online Update with Adaptive Population**   In the fixed population update rule, the number of the past individuals $n$ is fixed during the update. It means that every new individual added to the calculation will have the same influence in every update. To make this process more adaptive to each individual, we consider Rechenberg's method [34]. It makes its variance increase for better exploration when success rate is high, and decrease for more stability when success rate is low.

We design $n$ by utilizing the update ratio $p$ from the mean calculation in Sec. 3.3, to replace 'success rate' in [34] to be more suitable for our algorithm, as well as to smooth and stabilize the update process. $p$ implies how 'important' the new individual is.

To elaborate, assume $p$ of the newly evaluated individual is high. This means that the individual is relatively 'important' than the other individuals. We want to increase the influence of this individual in the variance update, by reducing the value of $n$, which reduces the influence of the previous variance $\sigma_{t-1}$ in Eq. (11). If the new individual is relatively not 'important', i.e. it has small $p$, then $n$ is set to a larger value. When $p = 0$, then the new individual has no importance, the variance remains unchanged. With a clipping operation added, we can express the update process of $n$ as

$$n = \max\left(\frac{1-p}{p}, 1\right) \tag{12}$$

Note that the mean and the variance update rules are independent, so any combination of the rules can be used.

Table 1: **Time efficiency comparison results in HalfCheetah-v2, Walker2d-v2 and Hopper-v2.**
In HalfCheetah-v2, the episode length is fixed to 1000 steps, and evaluation time for all agents are theoretically identical. In Walker2d-v2 and Hopper-v2, the episode length varies. The percentage indicates the amount of *reduced* time compared to CEM-RL. More detailed results about the number of workers are provided in Appendix F.

| | CEM-RL | P-CEM-RL | AES-RL | |
|---|---|---|---|---|
| Sequence Update | Serial Sync. | Parallel Sync. | Parallel Async. | |
| Workers | 1 | 5 | 5 | 9 |
| HalfCheetah-v2 (min) | 467 | 187 (59.9%↓) | 83 (82.1%↓) | 54 (88.3%↓) |
| Walker2d-v2 (min) | 487 | 205 (57.9%↓) | 105 (78.4%↓) | 77 (84.1%↓) |
| Hopper-v2 (min) | 505 | 188 (62.6%↓) | 133 (73.6%↓) | 91 (82.0%↓) |

Table 2: **Performance analysis for various combinations of our proposed mean-variance update methods.** Results are measured with average scores of ten test runs within a total of 1M step from the summation of all worker steps, averaged with ten random seeds. Full results for various environments are presented in Appendix D.

| | Category | Previous algorithms | | Proposed algorithms | | | | | | | |
|---|---|---|---|---|---|---|---|---|---|---|---|
| | | (1+1)-ES | Rank-Based | Fixed Range | | Fitness Baseline | | Fixed Range | | Fitness Baseline | |
| | $\mu$ | Full Move | Oldest | Linear | Sigmoid | Absolute | Relative | Linear | Sigmoid | Absolute | Relative |
| | $\Sigma$ | Success Rule | | Online Update - Adaptive | | | | Online Update - Fixed | | | |
| HalfCheetah-v2 | Mean | 11882 | 10010 | 10279 | 12053 | 12224 | **12550** | 10870 | 12031 | 11767 | 12128 |
| | Std. | 385 | 746 | 1044 | 398 | 422 | **187** | 409 | 604 | 458 | 821 |
| Walker2D-v2 | Mean | 2347 | 4230 | 5020 | 5360 | 5137 | **5474** | 3419 | 5121 | 5039 | 5070 |
| | Std. | 320 | 254 | 799 | 683 | 223 | **223** | 1676 | 965 | 471 | 557 |

# 4 Experiment

In this section, we compare the results of provided methods with the synchronous algorithm in the perspective of time-efficiency and performance. We evaluate the algorithms in several simulated environments which are commonly used as benchmarks in policy search: HalfCheetah-v2, Hopper-v2, Walker2d-v2, Swimmer-v2, Ant-v2, and Humanoid-v2 [36]. The presented statistics were calculated and averaged over 10 runs with the same configuration. As our aim is to show the efficiency of the asynchronous methods, we tried to use architecture and hyperparameters as similar as possible of those in CEM-RL [19]. We used TD3 [5] in RL part. Detailed architecture and hyperparameters for all methods are shown in Appendix A and C, respectively. [1]

## 4.1 Time Efficiency Analysis

We compare the time efficiency of three architectures, serial-synchronous, parallel-synchronous, and parallel-asynchronous. CEM-RL from the original author[2] is based on a serial-synchronous method. We implement parallel-synchronous version of CEM-RL, P-CEM-RL. Finally, AES-RL is our proposed parallel-asynchronous method. All three algorithms are evaluated in HalfCHeetah-v2, which has the fixed episode steps, and Walker2d-v2 and Hopper-v2, which has varying episode steps, in the same hardware configuration, two Ethernet-connected machines of Intel i7-6800k and three NVidia GeForce 1080Ti; a total of 24 CPU cores and 6 GPUs.

The results are shown in Tab. 1. It shows that parallelization brings significant time reduction compared to the serial methods. By simply switching from serial to parallel, time efficiency is greatly improved, reducing around 60% of total training time compared to the synchronous serial method. Additional efficiency of around 10-20% is achieved from asynchronism. We also scale workers up to nine, then 80-90% of the training time is reduced compared to the serial version. The result shows that the inefficiency of the synchronous methods is significant with varying episode length, as described in Fig. 1.

## 4.2 Performance Analysis of Mean-Variance Update Rules

We propose the asynchronous update methods including 4 mean and 2 variance update rules, as described in Sec. 3. Therefore, 8 combinations are available, and we thoroughly analyze their properties. Tab. 2 shows the comparison results, along with the previously proposed algorithms, (1+1)-ES [34], and asynchronous version of rank-based update [26]. All algorithms are evaluated in HalfCheetah-v2 and Walker-v2 environment, where former is easier and latter is harder, relatively.

As we expected in Sec. 3, (1+1)-ES shows the instability because of its extreme update rule. It successfully solved the HalfCheetah-v2, but failed in Walker-v2. In contrast to the (1+1)-ES, Rank-based method shows lower performance in both environment. They saturated at the lower score and was not able to explore further.

The fixed range algorithms show lower performance compared to the baseline algorithms, but are better than the rank-based method. Sigmoid is better than linear for all configuration, since it leverages more advantages from superior individuals. The baseline methods show higher performance and stability, showing higher mean and lower variance, consistently. Here, stability is defined as how consistent the results are for various random seeds with the std value per mean $\sigma/\mu$. In the absolute baseline method, it updates conservatively at the latter part of training because of the scaling effect mentioned in Sec. 2.3. Whereas, the relative baseline method effectively utilizes the information from superior individuals, thus showing the best results. In terms of the variance update, the adaptive method shows better performance, compare to the fixed population method. The fixed population method shows higher variance and lower mean, because it sometimes fails to explore enough compared to the adaptive method. As a result, the relative baseline with adaptive variance method shows the best results as expected. Detailed result for combinations are presented in Appendix D.

## 4.3 Performance Analysis of Other Algorithms

We compare our proposed algorithm to the previous RL and ERL algorithms: TD3, SAC [6], CEM, ERL and CEM-RL. The results are summarized in Tab. 3. Except for our results, we cite the results of the RL and ERL algorithms reported in [19]. Excluding the ERL, the other algorithms are trained for 1M steps. Training steps of the ERL is the same as what were given in the original paper [13]:

Table 3: **Performance analysis of TD3, SAC, CEM, ERL, CEM-RL, and AES-RL in six MuJoCo benchmarks.** Our algorithm outperforms the other methods in most of environments except for Ant-v2 and Swimmer-v2. Results of other algorithms are from their original report, except for Humanoid-v2. Improvements are compared to the baseline CEM-RL.

| Environment | Statistics | TD3 | SAC | CEM | ERL | CEM-RL | AES-RL | Improvement |
|---|---|---|---|---|---|---|---|---|
| Algorithm | | RL | | ES | | RL+ES | RL+ES | |
| Update Method | | - | | - | | Sync. | Async. | |
| HalfCheetah-v2 | Mean | 9630 | 11504 | 2940 | 8684 | 10725 | **12550** | |
| | Std. | 202 | 183 | 353 | 130 | 397 | **187** | **17.02 %** |
| | Median | 9606 | 11418 | 3045 | 8675 | 11539 | **12571** | |
| Hopper-v2 | Mean | 3355 | 3239 | 1055 | 2288 | 3613 | **3751** | |
| | Std. | 171 | 18 | 14 | 240 | 105 | **58** | **3.83 %** |
| | Median | 3626 | 3230 | 1040 | 2267 | 3722 | **3746** | |
| Walker2D-v2 | Mean | 3808 | 4268 | 928 | 2188 | 4711 | **5474** | |
| | Std. | 339 | 435 | 50 | 328 | 155 | **223** | **16.19 %** |
| | Median | 3882 | 4354 | 934 | 2338 | 4637 | **5393** | |
| Ant-v2 | Mean | 4027 | **5985** | 487 | 3716 | 4251 | 5120 | |
| | Std. | 403 | **114** | 33 | 673 | 251 | 170 | **20.44 %** |
| | Median | 4587 | **6032** | 506 | 4240 | 4310 | 5071 | |
| Swimmer-v2 | Mean | 63 | 46 | **351** | 350 | 75 | 161 | |
| | Std. | 9 | 2 | **9** | 8 | 11 | 100 | - |
| | Median | 47 | 45 | **361** | 360 | 62 | 128 | |
| Humanoid-v2 | Mean | 5496 | 5505 | - | 5170 | 5579 | **6136** | |
| | Std. | 187 | 108 | - | 130 | 228 | **444** | **9.98 %** |
| | Median | 5465 | 5539 | - | 5120 | 5673 | **6133** | |

2M on HalfCheetah-v2, Swimmer-v2 and Humanoid-v2, 4M on Hopper-v2, 6M on Ant-v2 and 10M on Walker2d-v2.

Our algorithm outperforms most of the other methods consistently, except for Ant-v2 and Swimmer-v2. As reported in CEM-RL, Swimmer-v2 is a challenging environment for all RL algorithms. However, our methods show better performance than TD3, CEM-RL in Swimmer-v2, because of better exploration. In detail, our algorithm gets two high scores and eight low scores out of 10 trials, in contrast to CEM-RL and TD3 that got low scores for all trials. The results for failed trials are **111.97 $\pm$ 24.10**, and **355.95 $\pm$ 1.95** for the succeeded trials. This shows that even in our 8 failed attempts, we outscore CEM-RL, while showing the comparable results in 2 succeeded trials.

The learning curves for each environment are presented in Fig. 3. It shows that the AES-RL can achieve better performance with less time and also within a fixed amount of steps.

## 5   Discussion

In this paper, we proposed the asynchronous parallel ERL framework to address the problems of previous synchronous ERL methods. We also provided several novel mean-variance update rules for updating the distribution of the population and analyzed the influence of each method. Our proposed framework and update rules were evaluated in six MuJoCo benchmarks, and showed outstanding results with 80% of time reduction with five workers and 20% of performance improvement at its best.

With our framework, it is possible to use any off-policy RL algorithms and many ask-and-tell ES algorithms, which can be used with up to 200k parameters. We remain this for the future work.

Figure 3: **Wallclock-wise Learning Curve** Curves are averaged with three random seed

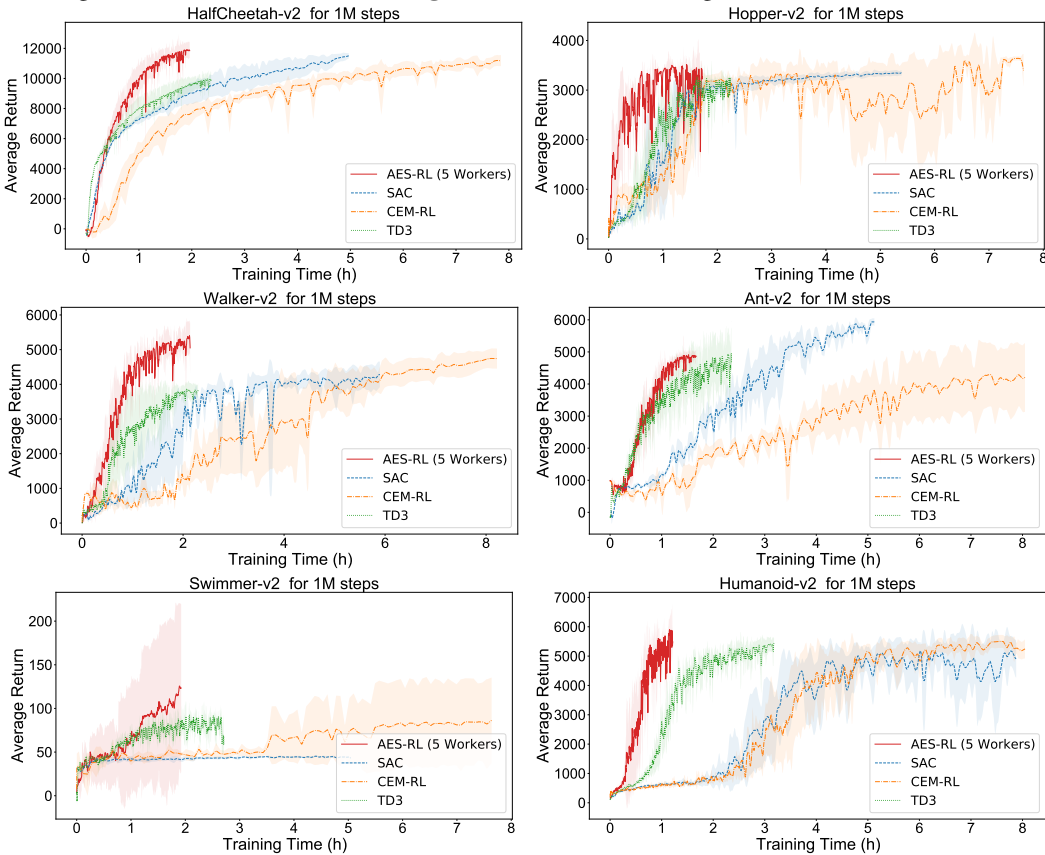

## Broader Impact

To apply RL in the real world problem, efficiency and stability are essential, because the cost of interacting with the environment is much higher than the simulation. Evolutionary Reinforcement Learning is an attempt to combine ES and RL for efficiency and stability. However, the previous works used synchronous evolutionary algorithms that cannot be scaled up.

This paper focuses on the asynchronous combination method of RL and ES. We introduce several efficient update rules for applying asynchronism and analyze the performance. We confirm that the asynchronous methods are much more time efficient, and it also have more exploration property for searching effective policy. With the proposed algorithm, AES-RL, branch of stable and efficient policy search algorithms can be extended to numerous workers. We expect the policy search algorithms to be actively applied to real world problems.

## Acknowledgments and Disclosure of Funding

This work was in partial supported by the Industrial Strategic Technology Development Program(Development of core technology for advanced locomotion/manipulation based on high-speed/power robot platform and robot intelligence, 1 0070171) funded By the Ministry of Trade, industry & Energy(MI, Korea) This work was also in partial supported by the Technology Innovation Program funded by the Ministry of Trade, Industry and Energy, South Korea, under Grant 2017-10069072.

## Footnotes

[1] The source code of our implementation is available at `https://github.com/KyunghyunLee/aes-rl`

[2] `https://github.com/apourchot/CEM-RL`

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
