[Supplementary Material]

## A  Network Architecture

For a fair comparison, our network follows the same structure as CEM-RL [19]. The architecture is originally from Fujimoto et al. [5], the only difference is using tanh instead of RELU. Pourchot and Sigaud [19] reported the difference between the RELU and tanh. We use $(400, 300)$ hidden layer for all environment except Humanoid-v2. For Humanoid-v2, we used $(256, 256)$ as in TD3 [5].

Table 4: **Network architectures** The architecture from the input layer to the output layer

| Layer Type | Actor | Critic |
|---|---|---|
| Linear | (state_dim, 400) | (state_dim + action_dim , 400) |
| Activation | tanh | leaky RELU |
| Linear | (400, 300) | (400, 300) |
| Activation | tanh | leaky RELU |
| Linear | (300, action_dim) | (300, 1) |
| Activation | tanh | |

## B  AES-RL pseudo-code

---

**Algorithm 1:** AES-RL

---

1 **Initialize:**  the mean of the population $\pi_\mu$, shared critic $Q^\pi$ and target critic $Q^{\pi'}$
2 **Initialize:**  the covariance matrix $\mathbf{\Sigma} = \sigma_{\text{init}}\mathcal{I}$, the emtpy replay buffer $\mathcal{R}$, $total\_steps = 0$
    /* Start training of the shared critic                                                */
3 critic_worker.start_critic_training()
4 **while** $total\_steps < max\_steps$ **do**
5     $N_{idle\_worker}$ = num_idle_worker()
6     **while** $N_{idle\_worker} > 0$ **do**
        /* Create new individual         */
7         new_individual = population.sample($\pi_\mu$, $\mathbf{\Sigma}$)
8         Create new individual by sampling from $\mathcal{N}(\pi_\mu, \mathbf{\Sigma})$
9         actor_worker = get_idle_worker()
10         actor_worker.set_actor_network(new_individual)
11         Calculate $p_{\text{rl}}$ according to Eq. (3)
12         **if** $rand() < p_{rl}$ **and** $total\_step >= rl\_start\_step$ **then**
            /* Train the actor with shared critic       */
13             critic_weight = critic_worker.get_critic_weight()
14             actor_worker.set_critic_network(critic_weight)
15             actor_worker.train_actor_network()
16             $n_{\text{rl}} \leftarrow n_{\text{rl}} + 1$
17         **else**
            /* Evaluate immediately          */
18             $n_{\text{es}} \leftarrow n_{\text{es}} + 1$
19         **end**
        /* Evaluate the individual, with filling the replay buffer      */
20         actor_worker.evaluate($\mathcal{R}$)
21         $N_{idle\_worker} \leftarrow N_{idle\_worker} - 1$
22     **end**
23     $N_{finished\_worker}$ = num_finished_worker()
24     **while** $N_{finished\_worker} > 0$ **do**
25         finished_worker = get_finished_worker()
26         fitness, individual, current_steps = finished_worker.get_evaluation_information()
        /* Update population with the finished individual      */
27         population.update(fitness, individual)
28         $total\_steps \leftarrow total\_steps + current\_steps$
29         $N_{finished\_worker} \leftarrow N_{finished\_worker} - 1$
30     **end**
31 **end**

---

**Algorithm 2:** actor.evaluate($\mathcal{R}$)

**1 Require:** hyperparameter action noise $a_{\text{noise}}$
**2** state = env.reset()
**3** done = False
**4** steps = 0
**5** total_reward = 0
**6 while** *not done* **do**
**7**     action = actor_network.forward(state)
**8**     **if** $a_{noise} \neq 0$ **then**
**9**        action = clip(action + $a_{\text{noise}}$ * random.normal(0, 1), -1, 1)
**10**     **end**
**11**     next_state, reward, done, info = env.step(action)
**12**     $\mathcal{R}$.append(state, next_state, reward, done)
**13**     steps $\leftarrow$ steps + 1
**14**     total_reward $\leftarrow$ total_reward + 1
**15**     state = next_state
**16 end**
**17** return total_reward, steps

## C   Hyperparameters

Most of hyperparameters are the same value as CEM-RL [19]. However, the size of replay buffer is modified to $2e5$, we analyzed the effect on Appendix C.1. Also, we used action noise of $0.1$ as suggested by Khadka and Tumer [13]. Pourchot and Sigaud [19] reported the action noise is not useful for their algorithm, we found that the action noise improves exploration as the original ERL paper. We use $a_{\text{noise}} = 0.1$ for all environments. We discuss the effect on Appendix C.2. The $K_{\text{rl}}$ in population control is set to 50. $p_{\text{negative}}$ for fixed range algorithms are set to 0. We use '1/5' for success rate of $(1+1)$-ES. Also, we use $p_{\text{desired}} = 0.5$ for all environment except Swimmer-v2. CEM-RL reported that RL algorithms provide deceptive gradients, therefore most of the RL algorithms fail to solve. Therefore we use $p_{\text{desired}} = 0.1$, which means that the population of the ES is 9 times larger than the RL. Consequently, algorithms are become more dependent on the ES part.

Other values related to the range are presented in Tab. 5. We noticed that the corresponding values are about 1/6 of the maximum reward. For example, the reward value reaches up to 12000 in HalfCheetah, then the 1/6 of the maximum is about 2000.

Table 5: **Hyperparameters** A list of the hyperparameters that vary with the environment

|  |  | Fixed Range | |
| --- | --- | --- | --- |
|  |  | Linear | Sigmoid |
|  | HalfCheetah-v2 | 2000 | 2000 |
|  | Hopper-v2 | 600 | 600 |
| Range $r$ | Walker2D-v2 | 860 | 860 |
|  | Ant-v2 | 960 | 960 |
|  | Swimmer-v2 | 48 | 48 |
|  | Humanoid-v2 | 960 | 960 |
|  |  | Fitness Baseline | |
|  |  | Absolute | Relative |
|  | HalfCheetah-v2 | -2000 | 2000 |
|  | Hopper-v2 | -600 | 600 |
| Baseline $f_b$ | Walker2D-v2 | -860 | 860 |
|  | Ant-v2 | -960 | 960 |
|  | Swimmer-v2 | -48 | 48 |
|  | Humanoid-v2 | -960 | 960 |

## C.1 Replay Buffer Size

The replay buffer size of 200k improves the performance of the proposed algorithm. The performance of CEM-RL also increased, but not as much as the asynchronous algorithm. A possible hypothesis for this phenomenon is as follows. In CEM-RL, the critic learning steps are guaranteed in the synchronous stage. However, in AES-RL, the critic is trained in a parallel with the other workers; thus samples should be more productive. With the reduced size of the replay buffer, it is filled with more recent steps and replace the oldest experiences which is nearly useless. Therefore, the samples are more informative.

Tab. 6 shows the comparison results. In Walker-v2, the size of the replay buffer does not significantly affect the performance of AES-RL. However, the performance gap is relatively more significant in HalfCheetah-v2. One of the differences of the two environment is that the learning saturates earlier in Walker-v2. Therefore, the replay buffer is productive enough. Differently, actors in HalfCheetah-v2 still learn at the end of 1M steps. Here, We may use other techniques like importance sampling to enhance the efficiency of a mini-batch sample; however, we remain it as future work.

Table 6: **Effect of the Replay Buffer Size**

| | $\mu$ $\Sigma$ | Relative Range Adaptive | | CEM-RL | |
|---|---|---|---|---|---|
| | Replay | 200k | 1M | 200k | 1M |
| HalfCheetah-v2 | Mean | **12550** | 11472 | 11515 | 10725 |
| | Std. | **187** | 467 | 203 | 354 |
| Walker2D-v2 | Mean | **5474** | 5468 | 4503 | 4711 |
| | Std. | **223** | 690 | 388 | 155 |

## C.2 Action Noise

CEM-RL reported that the action noise is not useful, as opposed to ERL. However it consistently improves the performance a little in our experiments. We hypothesize that the $\pi_\mu$ of CEM-RL moves with the weighted average of individuals; therefore, the effect of action noise is reduced. Otherwise, in AES-RL, the action noise increases the chance of better policy which affects directly to the $\pi_\mu$.

Table 7: **Effect of the Action Noise**

| | $\mu$ $\Sigma$ | Relative Range Adaptive | |
|---|---|---|---|
| | $a_{\text{noise}}$ | 0.1 | 0.0 |
| HalfCheetah-v2 | Mean | **12550** | 12095 |
| | Std. | **187** | 338 |
| Walker2D-v2 | Mean | **5474** | 5244 |
| | Std. | **223** | 670 |

# D  Full Results of Proposed Methods

We compare all combination of methods proposed in Sec. 3. Except for Swimmer-v2, the relative baseline is the best in both performance and stability. In Hopper-v2 and Ant-v2, the relative baseline scores are slightly lower than the best methods, but it is comparable. Therefore we use the relative baseline methods when compare with the previous algorithms: TD3, CEM, ERL, and CEM-RL.

In Swimmer-v2, CEM, pure evolutionary algorithm, was the best, ERL, mostly evolutionary algorithm, was also able to solve the environment [19]. Among the asynchronous algorithms, $(1 + 1)$-ES, which has the most aggressive update rule, is consistently successful. From this result we can infer that aggressive exploration is more important in Swimmer-v2. Also, other methods have a chance to solve the environment, but not always.

In conclusion, we proposed various update rules for asynchronous algorithms. We started from the Rank-Based asynchronous algorithm and the $(1 + 1)$-ES update algorithm, which are at the extremes.

Our design purpose is to achieve a balance between the two. The relative baseline method with adaptive variance showed the best performance, which effectively balances between aggressive and conservative updates.

Table 8: **Full results of our proposed mean-variance update methods** Results are measured with average scores of ten test runs within a total of 1M step from the summation of all worker steps, averaged with ten random seeds.

| Category | | Previous algorithms | | Proposed algorithms | | | | | | | |
|---|---|---|---|---|---|---|---|---|---|---|---|
| | | (1+1)-ES | Rank-Based | Fixed Range | | Fitness Baseline | | Fixed Range | | Fitness Baseline | |
| | $\mu$ | Full Move | Oldest | Linear | Sigmoid | Absolute | Relative | Linear | Sigmoid | Absolute | Relative |
| | $\Sigma$ | Success Rule | | Online Update - Adaptive | | | | Online Update - Fixed | | | |
| HalfCheetah-v2 | Mean | 11882 | 10010 | 10279 | 12053 | 12224 | **12550** | 10870 | 12031 | 11767 | 12128 |
| | Std. | 385 | 746 | 1044 | 398 | 422 | **187** | 409 | 604 | 458 | 821 |
| Walker2D-v2 | Mean | 2347 | 4230 | 5020 | 5360 | 5137 | **5474** | 3419 | 5121 | 5039 | 5070 |
| | Std. | 320 | 254 | 799 | 683 | 223 | **223** | 1676 | 965 | 471 | 557 |
| Hopper-v2 | Mean | 2588 | 3729 | 3506 | 3764 | **3789** | 3751 | 2996 | 3769 | 3788 | 3423 |
| | Std. | 748 | 53 | 179 | 40 | **26** | 58 | 1068 | 20 | 30 | 626 |
| Ant-v2 | Mean | 5098 | 3917 | 3015 | **5140** | 3883 | 5120 | 3406 | 5007 | 4947 | 4613 |
| | Std. | 715 | 514 | 1233 | **546** | 1047 | 170 | 944 | 891 | 543 | 712 |
| Swimmer-v2 | Mean | **347** | 59 | 99 | 128 | 97 | 161 | 191 | 81 | 107 | 120 |
| | Std. | **19** | 8 | 43 | 65 | 32 | 100 | 123 | 12 | 34 | 63 |
| Humanoid-v2 | Mean | 600 | 3476 | 5697 | 5958 | 5770 | **6136** | 5662 | 5695 | 5774 | 5837 |
| | Std. | 35 | 1980 | 177 | 301 | 229 | **444** | 107 | 209 | 267 | 239 |

# E  Ablation Study

AES-RL algorithm mainly consists of three novel methods; an asynchronism, the mean update rule, and the variance update rule. In this section, we evaluate the effectiveness of each methods.

## E.1  Asynchronism

To compare the effectiveness of asynchronism, we adopt an update rule of CEM-RL based on the simple asynchronous methods in [26]. Therefore, the resulting algorithm is an asynchronous version of CEM-RL, namely ACEM-RL. As in [26], previous results are stored in a separate list with its score. When a new individual is evaluated, it is stored on the list, and the oldest one is removed to maintain the population size. However, there should be a multiplication factor $1/n$ because the update occurs $n$ times frequently. Therefore the mean update in Eq. (1) is modified to

$$\mu_{t+1} = \frac{1}{n} \sum_{i=1}^{K_e} \lambda_i z_i \tag{13}$$

## E.2  Mean and Update Rule

In the mean update, we already compared various update rules in Sec. 4.2 and Appendix D. The result of ACEM-RL, which applies asynchronism to CEM-RL, is in column "Rank-based"-"Oldest" in Tab. 8 In addition, we fix the variance to the constant value. Here, We expect that the fixed variance prevents exploration.

The overall results are displayed in Tab. 9. As a result, simply applying asynchronism to the previous method without proper mean and variance update reduces the performance.

Table 9: **Ablation study** The overall result of the ablation study. From the baseline algorithm CEM-RL, ACEM-RL adopts asynchronism. AES-RL with constant variance means that only the mean is updated. Finally, AES-RL results include all features, asynchronism, mean update, and variance update.

| | | CEM-RL | ACEM-RL | AES-RL $\sigma^2$ = const. | | **AES-RL** |
|---|---|---|---|---|---|---|
| | | | | 0.0001 | 0.001 | |
| HalfCheetah-v2 | Mean | 10725 | 10010 | 11636 | 12306 | **12550** |
| | Std. | 397 | 746 | 209 | 233 | **187** |
| Walker2D-v2 | Mean | 4711 | 4230 | 5167 | 5302 | **5474** |
| | Std. | 155 | 254 | 251 | 458 | **223** |

# F    Training Time According to the Number of workers

We compare the execution time for a various number of workers, from 2 to 9. Training time of CEM-RL is measured with original author's implementation. P-CEM-RL is our implementation of parallel version of CEM-RL, which has parallelized actors with synchronous update scheme. For AES-RL,

Table 10: **Training time according to the number of workers** Training time is measured in minutes with Ethernet-connected two machines of Intel i7-6800k with three NVidia GeForce 1080Ti each.

|         | CEM-RL | P-CEM-RL | AES-RL | | | | | | | |
|---|---|---|---|---|---|---|---|---|---|---|
| Workers | 1 | 5 | 2 | 3 | 4 | 5 | 6 | 7 | 8 | 9 |
| HalfCheetah-v2 | 467.17 | 187.42 | 225.63 | 136.32 | 103.15 | 83.43 | 72.38 | 65.52 | 58.88 | **54.47** |
| Walker-v2 | 487.25 | 205.17 | 275.90 | 163.10 | 131.18 | 105.42 | 97.12 | 84.23 | 81.02 | **77.33** |
| Hopper-v2 | 504.63 | 188.48 | 305.23 | 199.55 | 144.77 | 133.23 | 122.58 | 97.35 | 92.52 | **90.75** |

# G    Contribution of RL and ES in Learning Process

We approximately compare the contribution of RL and ES agents. To measure the contributio,n we used the update ratio $p$ in mean update rules. Higher $p$ indicates the new agent moves the mean of the distribution more. We recorded the $p$ value for all updates, and the values are accumulated for total experiment. The result in Tab. 11 shows that the ratio of each agents differs in each environments. For Swimmer-v2, our agent fails to find good solution (reward higher than 300) 8 out of 10, we only measured in successful trials.

Table 11: **Contribution of RL and ES**

|        | HalfCheetah-v2 | Walker2D-v2 | Hopper-v2 | Ant-v2 | Swimmer-v2 | Humanoid-v2 |
|---|---|---|---|---|---|---|
| ES (%) | 37.1 | 50.3 | 64.2 | 22.5 | 79.5 | 50.4 |
| RL (%) | 62.9 | 49.7 | 35.8 | 77.5 | 20.5 | 49.6 |

# H    Step-wise Learning Curve