[Reviews · NeurIPS 2020]

Review 1

Summary and Contributions: The paper introduces AES-RL, an asynchronous framework to combine Evolutionary Strategies and off-policy policy gradient to enable parallelizability. Results in a host of continuous control benchmarks demonstrate that the proposed method improves over baselines in terms of runtime and final performance.

Strengths: 1) The idea is simple but powerful and well-motivated. Parallelizability is often hampered by heterogeneous episode runtimes in the population. The ability to run population updates asynchronously is thus a major advantage that can enable a host of applications. 2) Facilitating online updates by leveraging Welford and Rechenberg update rules is very innovative and seems to work fairly well. 3) The improvements in wall-clock time achieved from the method is significant. While the benchmarks presented in the paper are based on Mujoco, the wall clock gains would be significantly more when using higher-fidelity physics engines such as Opensim or Gazebo where the disparity between simulation steps can be much more pronounced. This can serve to be an enabling technology for application of these methods towards the use of high-fidelity physics simulators.

Weaknesses: 1) Humanoid is a major environment missing from the suite of benchmarks tested on. I would be curious to know how the method performs in Humanoid? 2) The paper is missing a targeted case study that dive into the detail of how the ES and PG workers in the population interact. For example, how useful are the ES-updates in comparison to the PG updates within the population? Does this differ for each task? It would be reasonable to assume so. Detailed experiments that elucidate the inner working of the method would greatly strengthen the claims of the paper.

Correctness: The proposed method seem technically sound. The algorithm is well-presented and the design choices seem justified.

Clarity: The paper is extremely well written and easy to follow. Illustrations such as in Figure 1 and 2 are extremely helpful in elucidating the proposed method. The appendix in particular provides an extremely detailed account of the method and the expeirments.

Relation to Prior Work: The paper does a good job of highlighting the relevant background in hybrid methods that combine EA and off-policy policy gradients. It provides enough context for a reader to be able to place the proposed method amongst the broader field.

Reproducibility: Yes

Additional Feedback: 1) The ideas being the fixed range update (with or without the baseline) is similar to trust region based updates popularized by TRPO and PPO. It would be great if the authors can draw comparisons with these well-established methods. Further, ideas from the wider literature surrounding trust-region based optimization could provide an avenue for further growth direction for the proposed method. 2) How does the method scle to discrete settings like Atari? Broad experiments that include these benchmarks would greatly increase the appeal of the proposed method to a broader audience of RL practitioners. ### POST REBUTTAL ### I have read the author's rebuttal and other reviewer's comments. Accordingly I maintain my original score recommending acceptance for the submission.


Review 2

Summary and Contributions: Deep reinforcement learning (DRL) and evolution strategies (ES) are recently combined to make up for their stability issue and sample inefficiency to each other. This paper proposes an asynchronous version of such a combination of DRL and ES to improve its parallel performance. The authors extend the previous work, CEM-RL, to support asynchronous update of a population of agents. Two issues in making the existing algorithm asynchronous are mentioned and addressed. Stated difficulties in extending CEM-RL to asynchronous setting are as follows. 1. "CEM-RL creates a fixed number of ES and RL actors in new generation only after the all actors are terminated. But in the case of the asynchronous update, it is impossible to use this method because the actors have different ending moments." 2. "Secondly, we need an asynchronous method to update population distribution effectively and also stably. Since each actor has its own fitness value, a novel update method to adaptively exploit this is required." These difficulties are addressed by introducing a control mechanism of the ratio between the populations of RL and ES actors, and by employnig a modified distribution parameter update formula. Experimental evaluation on Mujoco environments shows its speed-up by upto 25% compared to the parallel (synchronous) baseline and performance improvement over existing approaches. However, ablation studies are missing and it is unclear whether the performance improvement comes from the speed-up by asynchronization.

Strengths: 1. Stability and sample efficiency are important issues to be addressed in DRL, and a combination of ES and RL have been reported as a way to improve these difficulties. Asynchronous parallel implementation of such a method is a natural way to improve their wall clock time. The topic of this paper is suitable to NeurIPS. 2. Empirically, a speed-up (in wall clock time) and improvement in the final performance have been evaluted on mujoco environments over different SOTA algorithms, TD3, CEM, ERL, CEM-RL.

Weaknesses: 1. Asynchronization is introduced as a way to speed-up a parallel implementation of an algorithm with possibly compromizing sample efficiency (compared with synchronous version), rather than to improve the performance. Showing a performance improvement at a fixed wall-clock time or a fixed nubmer of interaction do not really show the goodness of asynchronization. What is the maximum time budget is higher or lower? Why not showing a performance graph? 2. Ablation study is missing. It is unclear where the performance improvement comes from. 3. As mentioned in the summary above, two difficulties in implementing asynchronous distribution update are stated in this paper. However, I can not agree with the first difficulty: "in the case of the asynchronous update, it is impossible to use this method because the actors have different ending moments." It is definitely not "impossible". The second difficulty is already addressed in asynchronous ES [25]. Therefore, it should be able to simply combine asyncronous ES with CEM-RL. I am not sure whether the proposed approach is really promising compared to this very simple baseline. It is also because of the lack of ablation study.

Correctness: It is unclear whether the performance comparison is done with the fixed wall-clock time or with the fixed interaction. The experimental details are missing. Some statements are not correct nor supported with evidence. "Fitness-based methods use the fitness value itself rather than rank, making it available to fully utilize the superior individual. (1+1)-ES [33] is one of the early work that uses fitness value in extremely aggressive way, ..." It is wrong. (1+1)-ES [33] does not use fitness-value itself, and it used only the comparison result of two fitness values. "In order to restrain the aggressiveness of (1+1)-ES, while preserving the capability of high exploration in fitness-based scheme, ..." Provide an evidence or reference for "the capability of high exploration in fitness-based scheme". It is rather counter-intuitive.

Clarity: Yes. The organization of the paper is clear.

Relation to Prior Work: In a global picture, the paper is well-positioned in related works. From a technical viewpoint, components introduced in Section 3.3 and 3.4 are tightly related to existing approaches ([7,25] and fitness shaping in natural evolution strategies) but not clearly stated.

Reproducibility: Yes

Additional Feedback: It is better to provide the code to reproduce the experiments. Since the contribution of this paper is the speed-up in wall clock time, it may heavily depend on the implementation. -- The author's response is satisfactory. Most of my concern are reflected.


Review 3

Summary and Contributions: The paper builds on the (interesting) recent trend of combining ES and RL. It adds significant improvements: - better asynchronous management. - new update rules for CEM in an ask-and-tell framework compatible with the asynchronous setting. The new update rules are actually not only for the asynchronous setting; they do provide improvements compared to CEM.

Strengths: Based on the combination RL/Evolution which looks quite cool. Looks like the approach does outperform interesting recent papers.

Weaknesses: A few issues can be solved for the final version (better caption for tables or a few more lines in the corresponding text). A bigger issue is that the many improvements associated to the update rules are not compared with other rules which are compatible with asynchronous generation of offspring.

Correctness: Yes.

Clarity: Yes.

Relation to Prior Work: Except for the other asynchronous methods. Maybe the authors can point out why existing methods presented in an ask-and-tell format, available open source and known as faster, are not suitable. The one-plus-one ES is definitely not fitness-based but comparison-based; I maintain my recommendation for acceptance.

Reproducibility: Yes

Additional Feedback: Sometimes rank-based black-box optimization is considered as more robust so some people might disagree with the claim that it's better to use of fitness values (as opposed to only the ranks). Regarding this usage of the detailed fitness values, there are many parameters, so the stability of the method could be discussed. The presentation of the state of the art (combinations of ES and grad-based policy search) is ok. The code of baselines is mentioned in the footnote; you might also provide your code (with github anonymizer), this does not break the double blind reviewing process. Remarks: - The ask and tell format becomes prevalent in black-box optimization; this allows asynchronous generation of new candidates. Sections 3.1 and 3.3 do not take into account this; generating a single point is feasible by many methods, more sophisticated than the (destructive) 1+1. - Table 1: maybe the caption can be more self-contained; how many different actors are running here ? The table suggests that you get only a factor 2 or 3 speed-up (which is not the case if I understand correctly). - Table 2: also more self-contained captions could help, or a bit more info immediately next to the caption; these results are with how many parallel actors and a fixed total time ? - Table 3: also more self-contained captions. Rebuttal: maybe provide answers for remarks above. You can probably easily answer points raised about tables 1, 2, 3. I think the existence of many ask and tell algorithms (for example CMA has an ask-and-tell version, and Nevergrad provides *all* its algorithms in ask and tell format) is a deeper remark. This does not mean that the paper should not be accepted (I'm not aware of anyone having done better than you for these problems so you can't blame for not doing every thing that could have been done!). CEM is not the only possibility, so many algorithms could be tried here. Detail (not taken into account for the overall evaluation) : the english is sometimes suboptimal (though always readable), e.g. mastering Go rather than "the Go".


Review 4

Summary and Contributions: This paper proposes a unique asynchronous framework for effectively combining Evolution Strategies (ES) and Deep Reinforcement Learning algorithms. The authors introduce various methods to minimize problems that may occur during the asynchronous update. Experimental results show that high performance can be obtained while reducing learning time through the proposed framework.

Strengths: - Various asynchronous update methods are proposed, and the characteristic of each method are well compared. - It is practically useful by increasing the efficiency of parallel training. - The baseline used in the experiment was thoroughly selected and compared. - The experimental results support the benefit of the proposed method.

Weaknesses: - This paper argues that the proposed asynchronous framework is more stable than the previous method. However, a metric to confirm stability is not defined, and experimental results do not support its stability also. - There is no analysis of performance and obstacle when using a variance matrix instead of a covariance matrix. - The experiment of time efficiency comparison was conducted only in a single setup. It would be helpful to have a comparison of different numbers of CPUs, GPUs, and workers.

Correctness: The empirical methodology seems correct. But some claims are not fully supported. The authors claim that the proposed method is more stable than the previous method, but stability cannot be confirmed in the paper.

Clarity: The paper is well written and easy to follow.

Relation to Prior Work: The main background is explained in detail, and the difference from the previous work is clearly described.

Reproducibility: Yes

Additional Feedback: #### POST REBUTTAL #### The authors' feedback covered my question. I recommend acceptance as in the previous review.

[Author Response · NeurIPS 2020]

We thank the reviewers for the reviews, providing meaningful insight with constructive feedback. Due to the page limitation, we only provide critical values. In the final manuscript, we will present all the results, alongside modifications reflecting reviewers' comments which are not mentioned in this response.

**R1: Result on Humanoid environment.** We tested our method on Humanoid-v2 and confirmed our method works properly. The relative baseline and adaptive variance update algorithm, which performs best among proposed algorithms, was tested on the environment and scored $\mathbf{6169 \pm 455}$ with ten random seeds at 1M steps.

**R1: Interaction of ES and PG workers.** We measured how many RL and EA actors were contributed in improving the performance, as a summation of the update ratio $p$ (Eq. 6), with higher value indicating more contribution. In our method, RL actors contributed twice more compared to ES actors in HalfCheetah, with values of **214.53** and **105.52**, respectively. The result was reversed in Hopper, where RL contributed **200.86** while EA actors did **363.53**.

**R2, R3: Evaluation method for performance and speed.** We evaluated our algorithm in two perspectives; performance improvement and speed improvement. For the performance improvement, we evaluated our method as same as the baselines for a fair comparison. In many papers, the final score of the fixed interaction step is frequently used for evaluation metrics. Therefore, all performance result scores are measured in the fixed interaction step. For the speed improvement, we measured execution wall-clock time for the fixed interaction step; the result is presented in Table 1. As the CEM-RL is implemented in a serial-synchronous, we modified the algorithm to a parallel-synchronous version (P-CEM-RL), and then we compared these algorithms with our method to show the efficiency of the asynchronism. We will include the learning curve in the final version.

**R2: Ablation study is missing.** Our algorithm mainly consists of three aspect; asynchronism, mean and variance update rules. We presented the effect of a simple asynchronous method [25] with the CEM-RL update rule in column "Rank-based" of Table 2. We presented the effect of the variance update rule in Appendix C.3 by comparing the result with a fixed variance setting. Then, we provided all combinations of our proposed mean and variance in Table 2. However, we agree with the reviewer because all these results are shown separately and not discussed thoroughly in the manuscript. We will add a section so that it can be seen at a glance. If these results are still not enough for an ablation study, it would be beneficial for us to consolidate our manuscript if the reviewer can provide us more specific guideline.

**R2: Asynchronism in CEM-RL is not "impossible".** We used the word "impossible" to emphasize that the update rule of CEM-RL cannot be used exactly the same in an asynchronous setting. CEM-RL spawns all actors at the same time with a fixed number of each agent. In an asynchronous setting, some modifications should be applied, such as alternatively spawning RL and ES actors. We intended to highlight the difference; however, we agree that the word is too aggressive. We will soften the tone in the final version.

**R2: (1+1)-ES is not a fitness-based method.** We categorized (1+1)-ES as a fitness-based method because it uses fitness values for comparing. However, as the reviewer pointed out, the method is ambiguous to be categorized. We will add detailed explanation in the final version.

**R2: No reference or evidence about the statement "Capability of high exploration in fitness-based scheme".** Our phrase "high exploration" is intended to empathize with the aggressiveness when there appears a superior individual. However, there might be a misunderstanding about the general meaning of exploration in the policy search field. We will modify the statement in the final version.

**R3, R4: Speed up is the factor of 2 or 3. Time efficiency experiment was conducted with only one setting.** We used five actors as provided in the section 4.1. The number of actors is not limited, but it requires GPU calculation, limiting the experiment in our setting. We additionally tested our methods with various actors of 2, 3, 4, 5, and 7 in Halfcheetah. The running times were **75%**, **42%**, **37%**, **32%**, and **25%** compared to the execution time of the CEM-RL. It shows that time efficiency is linearly increased as number of actors increases. We will provide broader experiments and discussions about the number of actors, with a table and also a graph.

**R3, R4: Ask and Tell based update rules are missing. Using variance instead of covariance.** We took a look into the Nevergrad library and read the original papers of implemented algorithms. We will try to merge various algorithms that fit the update scheme of combining ES and RL. It seems though some algorithms are hard to be merged. Our network consists of only three layers with 100k parameters, which is very shallow compared to the networks in the computer vision field. However, algorithms that use covariance like CMA-ES are not appropriate with 100k parameters. It is also the reason why the baseline CEM-RL used variance instead of covariance.

**R4: Stability metric is not provided.** We defined the term "stability" for consistently showing high performances, thereby having low std value per mean ($\sigma/\mu$). Our final method AES-RL, relative baseline with the adaptive update, is chosen because its performance is high, and the $\sigma/\mu$ value is low. As shown in Table 2, previous algorithms for asynchronous updates show a higher $\sigma/\mu$ compared to our method, therefore we claimed that our method is relatively stable. We will explicitly explain the metric in the plain text and also emphasize it in the tables.

[Meta-Review · NeurIPS 2020]

This paper provides a simple but effective approach to speeding up an integrated DRL and ES search, incorporating a novel application of previously proposed asynchronous updating rules and presenting experimental results that convincingly show the efficacy of the approach. The paper could use a better justification for its "greater stability" claim, could benefit from some comparison against competing asynchronous update rules, and has a few miscellaneous presentation problems that need to be addressed. But overall this paper makes a solid contribution and the recommendation is to accept. Please take care to address the reviewer comments as you prepare your final version.